# Features of the Carrier Concentration Determination during Irradiation of Wide-Gap Semiconductors: The Case Study of Silicon Carbide

**DOI:** 10.3390/ma15238637

**Published:** 2022-12-03

**Authors:** Alexander A. Lebedev, Vitali V. Kozlovski, Klavdia S. Davydovskaya, Roman A. Kuzmin, Mikhail E. Levinshtein, Anatolii M. Strel’chuk

**Affiliations:** 1Ioffe Institute, Politekhnicheskaya Street 26, St. Petersburg 194021, Russia; 2Department of Experimental Physics, St. Petersburg State Polytechnic University, Polytekhnicheskaya 29, St. Petersburg 195251, Russia

**Keywords:** radiation compensation, wide-gap semiconductors, deep levels, protons, electrons, current-voltage characteristics, capacitance-voltage characteristics

## Abstract

In this paper, the features of radiation compensation of wide-gap semiconductors are discussed, considering the case study of silicon carbide. Two classical methods of concentration determination are compared and analyzed: capacitance-voltage (*C*–*V*) and current-voltage (*I–V*) characteristics. The dependence of the base resistance in high-voltage 4H-SiC Schottky diodes on the dose of irradiation by electrons and protons is experimentally traced in the range of eight orders of magnitude. It is demonstrated that the dependence of the carrier concentration on the irradiation dose can be determined unambiguously and reliably in a very wide range of compensation levels, based on the results of measuring the *I–V* characteristics. It is shown that the determination of the carrier removal rate using the *I–V* characteristics is more correct than using the *C–V* characteristics, especially in the case of high radiation doses.

## 1. Introduction

The study of the wide-gap materials properties is one of the most dynamically developing areas of semiconductor physics. As a rule, materials with a band gap (*E*g) > (2.5–3.0) eV are referred to wide-band semiconductors. Compared to the classic semiconductor materials, Si and GaAs, wide-gap semiconductors make it possible to create radiation-resistant devices, operating at significantly higher temperatures [1,2,3]. Such devices can be used to improve the reliability of nuclear power plants, thermonuclear power plants already under design, and space technology devices that require the use of radiation-resistant semiconductor electronics. Such electronics are the subject of increased requirements for preserving the initial properties (and/or changing properties within acceptable limits) under the effect of various radiation types: protons, electrons, neutrons, alpha and gamma particles, as well as heavy high-energy particles.

Despite numerous studies in this area, many unresolved issues remain in the choice of options for the radiation resistance of wide-band semiconductors and in establishing possible ways to improve it. This paper considers one of the most important parameters characterizing radiation resistance: the change in the carrier concentration depending on the level of sample compensation due to irradiation with electrons and protons.

## 2. Statement of the Problem

To estimate the radiation resistance of semiconductors at a relatively low level of compensation, the parameter *η_e_* is often used, that is, the removal rate of carriers under the irradiation influence (see, for example, [4,5]):*η_e_* = (*n*_0_ − *n*)/*Φ*,(1)
where *n*_0_ is the electron concentration in the semiconductor before irradiation, *n* is the electron concentration after irradiation, and *Φ* is the fluence (for certainty, we will consider the n-type conductivity material).

In wide-gap semiconductors (i.e., SiC, GaN, AlGaN, Ga_2_O_3_), at the current level of technology, there are always deep centers (levels) that are practically not ionized at room temperature. When heated, the degree of their ionization increases, which leads to a temperature dependence of the measured *η_e_* value [6]. It should also be noted that, in determining the *η_e_* value, the concentration of uncompensated donors, *N_d_* − *N*_a_, is often used instead of the *n* value. In current *n*-type device-quality SiC, prior to irradiation, the concentration of carriers at room temperature can be considered equal with acceptable accuracy to the concentration of shallow donor *N_d_* (nitrogen). Irradiation with protons and/or electrons creates acceptor-type levels in the band gap [7,8]. As long as the total concentration of acceptor levels created by irradiation, $N_{a}^{\Sigma }$, is less than the initial carrier concentration *n*_0_, the residual electron concentration in the conduction band can be considered equal to *n* = *n*_0_ − $N_{a}^{\Sigma }$. However, at $N_{a}^{\Sigma }$ > *n*_0_, the shallow donor level responsible for the initial electron concentration in the non-irradiated semiconductor turns out to be completely empty, and the “residual” electron concentration *n* is determined by the generation of electrons from deep levels.

The *η* value can critically depend on the method of measurement. In the case of measuring current-voltage (*I–V*) characteristics or the Hall effect, the electron concentration after irradiation *n* is directly determined from the measurements. The value of *n* is determined directly, in this case, at any levels of compensation, including large fluences *Φ*, when expression (1) becomes inapplicable. When measuring capacitance-voltage characteristics (*C–V* measurements), carried out on reverse-biased Schottky diodes or p-n junctions, the *N_d_ − N_a_* value is measured. In this case, the removal rate *η_N_* (also at a relatively low level of compensation) is defined as:*η_N_* = [(*N_d_* − *N_a_*)_0_ − (*N_d_* − *N_a_*)_1_]/*Φ*(2)
where (*N_d_* − *N*_a_)_0_ is the initial value of the concentration of uncompensated donors, and (*N_d_* − *N*_a_)_1_ is the (*N_d_* − *N*_a_) value after irradiation.

Properly speaking, the degree of deep centers filling in the *I–V* (or the Hall effect) measurements and *C–V* measurements is always different. The quasi-neutral situation is realized when measuring the *I–V* characteristics. Then, the filling of the level is determined by two processes: thermal emission of electrons from the corresponding center, and recapture from the conduction band (*C*-band) to the level. If the level at the measurement temperature lies above the Fermi level, then it is empty; below it, it is filled.

In *C–V* measurements, the levels at which electrons are captured as a result of irradiation are located in the space charge layer, where there are practically no electrons in the *C*-band, and the carrier concentration at the level is determined by thermal emission from the level. At long measurement times, all levels lying in the upper half of the band gap should be emptied. However, for sufficiently deep levels, the time for complete depletion can be extremely long and significantly exceed the practically acceptable measurement time.

If the acceptor level is formed during irradiation in the lower half of the band gap, it will be filled with electrons, charged negatively, and will contribute to both a decrease in the *N_d_ − N_a_* value and a decrease in the concentration *n*.

If an acceptor level is formed in the upper half of the band gap, then it will reduce the *n* value. However, the decrease in the value of *N_d_* − *N*_a_ will depend on its depth (ionization energy). If the level ionization energy is large enough (deep level), then in a reasonable time for *C–V* measurements, it can remain almost completely filled with electrons. In this case, this level will contribute to the measured *N_d_* − *N*_a_ value. However, a relatively shallow level in *C–V* measurements can be completely emptied in a time much shorter than the measurement time already at room temperature. In this case, such a level will not contribute to the measured *N_d_* − *N*_a_ value.

Measurements of current-voltage and *C–V* characteristics of SiC Schottky diodes irradiated with protons and electrons were made in the Ref. [9]. Both types of irradiation lead to an increase in the resistance of the diode base. With large radiation doses, the resistance of the sample base increases extremely sharply, and measuring the *I–V* characteristics allows to trace the change in electron concentration by many orders of magnitude.

The value of *N_d_* − *N*_a_, determined from the capacitance–voltage characteristics at the same irradiation level, decreases at a much weaker rate [10,11]. In addition, since levels with different ionization empty at different rates, the measured *C–V* characteristics significantly depend on the measurement frequency and are characterized by a noticeable frequency dispersion at a given irradiation level [11].

The above considerations can be clearly illustrated using a simple qualitative model.

## 3. Qualitative Model

It is known that a number of deep levels are formed in silicon carbide under the actions of electron and proton irradiation [9,10,11,12,13,14,15]. Within the framework of the problem under consideration, these levels can be divided into three groups. Next, we will consider a simple model where each of these groups will be presented by one most characteristic level (Figure 1):

1. Shallow donor level D with *N_d_* concentration that existed in the sample before irradiation. The concentration of this level does not depend on the radiation dose. An impurity nitrogen level with an ionization energy *E*_i_ = 0.1 eV is considered such a level. With a sufficiently large fluence *Φ*, this level will be almost completely depleted in the quasi-neutral region. In the space charge layer, the level is completely empty at any *Φ*.

2. Relatively shallow acceptor level A1 with a concentration of *N*_A1_. Its characteristic relaxation time to the equilibrium value, $\tau =\frac{\tau_{e}\tau_{c}}{\tau_{e}+\tau_{c}}$, is determined by the time of electron emission from the level to the *C*-band *τ_e_* and the capture time *τ_c_*. Such a level will be filled in the quasi-neutral region and emptied in the space charge layer. It is expedient to choose the Z_1/2_ level with the ionization energy *E*_i_ = 0.68 eV, as such a model level is in 4H-SiC [5,8]. The appearance of this level as a result of irradiation will reduce the electron concentration in the conduction band *n*, determined from the current–voltage characteristics. However, as will be shown below, during *C–V* measurements, this level will be completely empty and not affect the decrease in the *N_d_* − *N*_a_ value.

3. The A2 level is located in the upper half of the band gap, but its ionization energy (*E_c_* − *E*_A2_) is quite high (>1 eV), so that during the characteristic time of measuring the *C–V* characteristics, the ionization of electrons from this level can be neglected. It is expedient to choose the E_6/7_ level with the ionization energy *E*_i_ ~ 1.5 eV, as such a model level in 4H-SiC [5,8].

4. Level A3 is located in the lower half of the band gap, and its ionization energy is even higher. The total concentration of very deep levels will be denoted as *N*_A_ = *N*_A2_ + *N*_A3_. These levels will decrease both the *n* value and the *N_d_* − *N*_a_ value.

Due to the large difference in the ionization energy of the levels under consideration, the concentration of electron *n_j_*, supplied to the conduction band from each level, can be considered with good accuracy independent of the filling of all other levels. Then (see, for example, [16]):(3)$$n_{j}=\frac{2(N_{d}-N_{a})}{1+\frac{gN_{a}}{N_{c}}\exp\varepsilon_{d}+{\left[{\left(1+\frac{gN_{a}}{N_{c}}\exp\varepsilon_{d}\right)}^{2}+\frac{4g(N_{d}-N_{a})\exp\varepsilon_{d}}{N_{c}}\right]}^{1/2}}$$
where *N_c_* is the density of states in the conduction band, $\varepsilon_{d}=\frac{E_{c}-E_{t}}{kT}$ (here, $E_{c}-E_{t}$ is the ionization energy of the corresponding level), and *g* is the degeneracy factor of the corresponding level. In calculations, the *g* was taken to be equal to unity (*g* = 1).

At low radiation doses (*N_d_* ≥ *N_A_*_1_ + *N_A_*), we took the total electron concentration *n* in the conduction band as the sum *n*_1_ + *n*_2_, where *n*_1_ and *n*_2_ are the concentrations of electrons delivered to the conduction band from levels *D* and *A*_1_, respectively, calculated by Formula (3).

When calculating the *n*_1_ according to Formula (3), the *N_d_* − *N*_a_ will obviously be equal to *N_d_* − *Φ*(a_A1_ + a_NA_), where a_A1_ is the rate of center *A*_1_ generation; a_NA_ is the total rate of all other acceptor levels’ generation. Note that the A1 level plays the role of an electron source in the *C*-zone, especially at high temperatures [17]. When calculating the value *n_j_ = n*_2_ according to (3), the *N_d_* − *N*_a_ will be equal to a_A1_*Φ*.

At high radiation doses (*N_d_* < *N_A_*_1_ + *N_A_*), the *D* level is completely empty. The residual conductivity of the base is due to the carrier’s ionization from level A1. Note that, in accordance with the adopted model, the probability of ionization from levels with higher ionization energies (A2 and A3 levels) is assumed to be zero.

In calculations, the rate of the center Z_1/2_ generation, determined from the analysis of DLTS spectra, was used.

Irradiation with 15 MeV energy protons was carried out in a portable MGTs-20 cyclotron. Irradiation with 0.9 MeV electrons was carried out on a resonant transformer accelerator. The electron beam’s current density was 12.5 µA cm^−2^. The pulse duration was 330 µs, and the pulse repetition rate was 490 Hz.

The carrier removal rates and generation rates of radiation defects used in the calculations are presented in Table 1.

## 4. Results and Discussion

Let us consider the situation using an example of irradiation with protons and electrons of 4H-SiC Schottky diodes with a blocking voltage of 600 V. The initial concentration of donors in the base, that is, the concentration corresponding to the *D* level, *N_d_*, is taken to be equal to 7 × 10^15^ cm^−3^ [6]. The density of states, *N_c_* at room temperature in the 4H-SiC is *N_c_* ≈ 9 × 10^18^ cm^−3^. Electron mobility *μ_e_* is assumed to be independent of fluence *Φ* and equal to 800 cm^2^/Vs [18].

Figure 2 presents the experimental dependences of the diode base resistance *R*_b_ = L/(enµS) on the doses with electron (Figure 2a) and proton (Figure 2b) irradiation [11,17]. Here, *e* is the electron charge, *n* is the electron concentration in the conduction band calculated in accordance with (3), *L* = 10 μm, and *S* = 6.75 × 10^−4^ cm^−2^ is the base length and diode area, respectively. These dependences are compared with the results of calculations in accordance with Equation (3).

Taking into account the fact that the change in the resistance *R*_b_ was traced in Figure 2 by eight orders of magnitude, the agreement between the experimental data and the calculations should be recognized as satisfactory. The obtained experimental results qualitatively agree with the data obtained in [7] during irradiation of 4H-SiC structures with an initial electron concentration of 7.2 × 10^15^ cm^−3^ by electrons with an energy of 400 keV.

Two characteristic parts can be distinguished in Figure 2.

At radiation doses of *Φ* ≤ 3 × 10^16^ cm^−2^ (electrons) and *Φ* ≤ 6 × 10^13^ cm^−2^ (protons), the resistance *R*_b_ increases relatively weakly (and practically linearly) with the radiation dose growth. This situation corresponds to the condition *N_d_* ≥ *N*_A1_ + *N*_A_. Conductivity is due to electrons ionized from the shallow donor level D. In this case, the carrier removal rates calculated by Formulas (1) and (2) will be close.At radiation doses *Φ* ≥ 3 × 10^16^ cm^−2^ (electrons) and *Φ* ≥ 6 × 10^13^ cm^−2^ (protons), there is a sharp increase in the base resistance with increasing *Φ*. The shallow level D is almost completely compensated, and the residual conductivity is due to the generation of electrons to the conduction band from the A1 level. (In more accurate calculations, it should be taken into account that, in addition to the Z_1/2_ level, which is presented in the model by the A1 level, electron irradiation also creates shallow levels with ionization energies of 0.25, 038, and 0.43 eV [5]).

It should be emphasized that the type of dependences *R*_b_(*Φ*), presented in Figure 2a,b, is of a general nature and characteristic of all studied SiC Schottky diodes.

As mentioned above, when measuring the *C–V* characteristics of reverse-biased Schottky diodes and p-n junctions, it is not the carrier concentration *n* that is measured, but the *N_d_* − *N*_a_ value. In this case, when calculating the value of *n* from the data of such measurements, it is assumed that all acceptor centers created by irradiation with a concentration of *N*_a_ remain negatively charged in the process of measuring the *C–V* characteristics. However, when a reverse voltage is applied, the acceptor centers in the space charge region become empty (neutral), and do not contribute to the measured *N_d_* − *N*_a_ value. In this case, the characteristic time of electron emission *τ_e_* from the corresponding level after the application of the reverse voltage is of fundamental importance. This time can be estimated as [19]:*τ_e_* = exp[(*E_c_* − *E_t_*)/*kT*]/*σν_t_N_c_*(4)
where *σ* is the capture cross-section, *ν_t_* is the electron thermal velocity, and *N_c_* is the density of states in the conduction band.

Taking for the Z_1/2_ level (level A1 in Figure 1) (*E_c_* − *E_t_*) = 0.68 eV, *σ* = 6 × 10^−14^ cm^2^, *v*_T_ ≈ 2 × 10^7^ cm/s, and *N_c_* = 1.7 × 10^19^ cm^−3^ [5, 18), we get *τ_e_* ≈ 4 × 10^−2^ s. For shallower levels with ionization energy 0.25, 038, and 0.43 eV [5], the time of depletion will be even shorter. Such times are significantly shorter than the practical time of measurement of the *C–V* characteristics. Thus, the value of *N_d_* − *N*_a_ obtained from *C–V* measurements is not, strictly speaking, equal to the decrease in the electron concentration *n* as a result of irradiation.

Note that for a deeper level A2 with an ionization energy *E*_i_ ~ 1.5 eV (Figure 1), identified with the E_6/7_ level [5,8], the emission time will be ~5 × 10^12^ s, which is many orders of magnitude greater than any practically acceptable time for measuring *C–V* characteristics. Thus, the E_6/7_ level and, of course, the levels lying in the lower half of the band gap will contribute to the measured *N_d_* − *N*_a_ value.

When estimating the *n* value from *C–V* measurements, a factor that makes it difficult to unambiguously interpret the experimental results is also a noticeable frequency dispersion of the capacitance of reverse-biased junctions in SiC structures.

Figure 3 shows the frequency dependence of the capacitance of a reverse-biased high-voltage 4H-SiC Schottky diode (CPW4-1200S002B diode with 1200 V blocking voltage), measured in the Ref. [11] over a wide frequency range.

As can be seen from Figure 3, the frequency dispersion in a non-radiated diode is rather insignificant, and it can be neglected at frequencies *f* ≥ 10 Hz. As the dose (fluence) *Φ* increases, the dispersion increases greatly, reflecting the appearance of relatively shallow centers, rapidly emptying when reverse bias is applied. The frequency dependence of the capacitance is due to the fact that at high frequencies, the deep centers do not have time to ionize, and at low and high frequencies, they find themselves in different charge states, giving a different contribution to the value of the measured capacitance.

It is important to emphasize that when measuring the carrier concentration from *I–V* characteristics, the directly measured value (base resistance *R*_b_) at an irradiation dose *Φ* = 7 × 10^16^ cm^−2^ changes with respect to *R*_b_ in a non-irradiated structure by ~5 orders of magnitude (Figure 2a). With *C–V* measurements, the capacitance under the same conditions only changes several times (Figure 3). It is obvious that the determination of the concentration from the *I–V* characteristics has an incomparably higher resolution.

## 5. Conclusions

In this study, the features of wide-band semiconductors radiation compensation were considered using the example of silicon carbide. The base resistance dependence of the high-voltage (blocking voltage 600 V) 4H-SiC Schottky diodes on the dose of irradiation with electrons (energy 0.9 MeV) and protons (energy 15 MeV) in the range of eight orders of magnitude was experimentally traced. It has been shown that the observed experimental dependences can be qualitatively interpreted on the basis of a simple three-level model that takes into account the formation of acceptor levels during SiC irradiation in the upper half of the band gap. It has been demonstrated that the measurement of current–voltage (*I–V*) characteristics makes it possible to uniquely and reliably determine the dependence of the carrier concentration (base resistance) on the radiation dose over a very wide range. It has been shown that determination of the carrier concentration from measurement of *I–V* characteristics has an incomparably higher resolution than *C–V* measurements.

## Figures and Tables

**Figure 1 materials-15-08637-f001:**
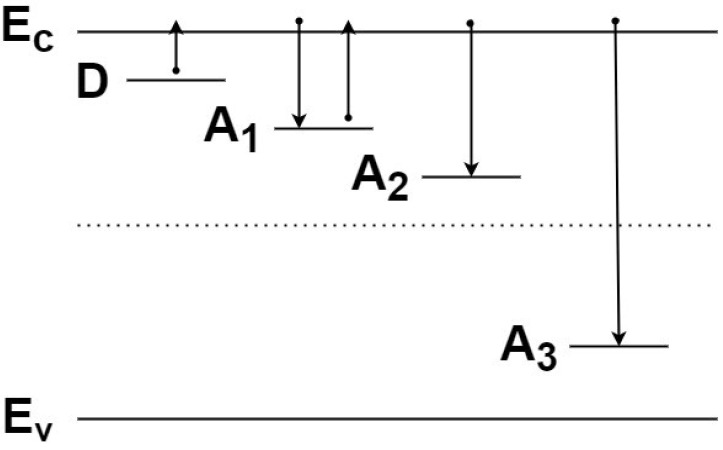
Schematic presentation of the main types of levels in the band gap of SiC. The arrows show the dominant electron transitions corresponding to the level. The dotted line corresponds to the middle of the band gap. Here, D is the shallow donor level; a relatively shallow acceptor level A1 associates with the Z_1/2_ level; an A2 level associates with the E_6/7_ level; and the A3 level represents the group of levels lying in the lower half of the band gap.

**Figure 2 materials-15-08637-f002:**
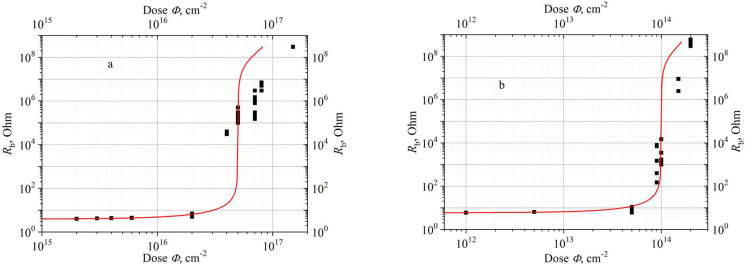
Experimental [9] and calculated (Equation (3)) dependences of the base resistance of 600 V 4H-SiC Schottky diodes, *R*_b_, on the dose *Φ* when irradiated with electrons with an energy of 0.9 MeV (**a**) and protons with an energy of 15 MeV (**b**). Points present experimental data, lines correspond to calculations.

**Figure 3 materials-15-08637-f003:**
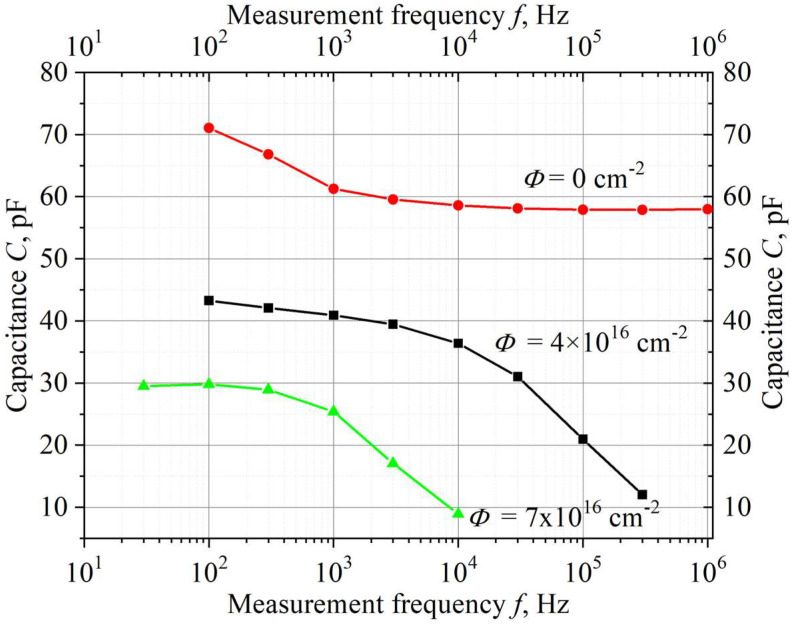
Frequency dependences of 4H-SiC Schottky diode capacitance at reverse bias *V* = −10 V for different values of fluencies *Φ* (irradiation with 0.9 MeV electrons).

**Table 1 materials-15-08637-t001:** Generation rates of deep levels.

	Electron Irradiation	Proton Irradiation
a_NA_ = *η_N_*, cm^−1^	0.14	70
a_Z1/2_, cm^−1^	0.034	13.2

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
