# Peer review of "Features of the Carrier Concentration Determination during Irradiation of Wide-Gap Semiconductors: The Case Study of Silicon Carbide"

_materials, 2022, doi:10.3390/ma15238637_

Round 1

Reviewer 1 Report

Manuscript ID: materials-2037911

A.A. Lebedev et al. have carried out radiation on SiC and experimentally traced in the range of 8 orders magnitude. The carrier concentration determination was compared and analyzed using volt-capacitive (C-V) and volt-ampere (I-V) characteristics. They have shown that the determination of the carrier removal rate using the I-V characteristics is more accurate than using the C-V characteristics, especially in the case of high radiation doses. Over all manuscript looks promising. However, I found some of the details are missing and should be included before publication.

Comments:

[1] Silicon carbide (SiC) exists in different polytypes (3C-SiC, 2H-SiC, 4H-SiC, 6H-SiC, R-15 and so on), which polytype considered for this study?

Please provide details in experimental section.

[2] What is the radiation source used to produce electrons and protons for this study?

[3] During trace of 8 order magnitude, is there any degradation observed?

[4] Additional characterization like photoluminescence could provide more insights into decay with radiation doses. I recommend authors to perform this characterization test.

Author Response

Dear Sir!

Thank you very much for the analysis of our thesis and the comments made. We have corrected the text in accordance with your comments. Changed parts of the text are highlighted in blue font.

[1] Silicon carbide (SiC) exists in different polytypes (3C-SiC, 2H-SiC, 4H-SiC, 6H-SiC, R-15 and so on), which polytype considered for this study? Please provide details in experimental section.

Thank you very much for this remark. We pointed the polytype (4H-SiC) in the text (pp. 1,5,6,7,8,9).

[2] What is the radiation source used to produce electrons and protons for this study?

To meet this Referee’s remark we have added the following text (p. 6):

Irradiation with 15 MeV energy protons was carried out in portable MGTs-20 cyclotron. Irradiation with 0.9 MeV electrons was carried out on a resonant transformer accelerator. The electron beam current density was 12.5 µA cm-2. The pulse duration was 330 µs, the pulse repetition rate was 490 Hz.

[3] During trace of 8 order magnitude, is there any degradation observed?

We would like to respectfully note that with such a huge increase in resistivity, the very concept of degradation becomes extremely complex and can only be determined in relation to a well-defined set of parameters. An investigation of this kind is beyond the scope of this work.

[4] Additional characterization like photoluminescence could provide more insights into decay with radiation doses. I recommend authors to perform this characterization test.

We fully agree with the Referee that the study of photoluminescence can provide significant additional information about structural changes and the nature of defects caused by irradiation. This kind of research may be the subject of future studies.

Reviewer 2 Report

Reviewer report:

The manuscript “Features of the carrier concentration determination during irradiation of wide-gap semiconductors (on the example of SiC)” is an interesting work in the field of radiation resistance of semiconductors. Main question addressed is determination of radiation influence of carrier concentration using I-V characteristics of Schottky diode. It is proven that I-V characteristics have advantages over C-V characteristics when determining carrier concentration. Paper is well written; text is clear and easy to read and is arranged in logical sequence. Conclusions address the main question. 

I recommend to publish manuscript after addressing following comments:

1 The title includes text in parenthesis which is unfamiliar. Better edit the title. Title and subtitles ending with punctuation marks? Check formatting of power (of Nc) in the beginning of section 4.

2 How base resistance is determined from I-V characteristics? Is it differential resistance or V/I. Is it measured at V=0 or some other point?

Author Response

Dear Sir!

Thank you very much for the analysis of our thesis and the comments made. We have corrected the text in accordance with your comments. Changed parts of the text are highlighted in blue font.  

1 The title includes text in parenthesis which is unfamiliar. Better edit the title. Title and subtitles ending with punctuation marks? Check formatting of power (of Nc) in the beginning of section 4.

The title includes text in parenthesis which is unfamiliar. Better edit the title.

It’s done. Thank you very much.

Title and subtitles ending with punctuation marks?

It’s corrected. Thank you very much.

Check formatting of power (of Nc) in the beginning of section 4.

It’s done. Thank you very much.

2 How base resistance is determined from I-V characteristics? Is it differential resistance or V/I. Is it measured at V=0 or some other point?

To clarify these points, we added at the beginning of section 4 references to papers 11 and 17, which contain detailed answers to these questions:

Fig. 2 presents the experimental dependences of the diode base resistance

1 The title includes text in parenthesis which is unfamiliar. Better edit the title. Title and subtitles ending with punctuation marks? Check formatting of power (of Nc) in the beginning of section 4.

The title includes text in parenthesis which is unfamiliar. Better edit the title.

It’s done. Thank you very much.

Title and subtitles ending with punctuation marks?

It’s corrected. Thank you very much.

Check formatting of power (of Nc) in the beginning of section 4.

It’s done. Thank you very much.

2 How base resistance is determined from I-V characteristics? Is it differential resistance or V/I. Is it measured at V=0 or some other point?

To clarify these points, we added at the beginning of section 4 references to papers 11 and 17, which contain detailed answers to these questions:

Fig. 2 presents the experimental dependences of the diode base resistance  on the doses with electrons (Fig. 2a) and protons (Fig. 2b) irradiation [11,17].

Reviewer 3 Report

This work by Lebedev is very minimalistic and dry but solid and the experimental studies behind the two figures is not negligible

the authors basically address the issue of determining carrier concentration upon irradiation and report wide dataset of R vs irradiation dose as well as C vs freq

the experimental results are accounted for  using a basic three-level model involving  the formation of acceptor levels during SiC irradiation in the upper half of the band gap

in general I would suggest to expand a bit the argumentations and possibly show more experimental data supporting the results, but here the target is well and sharply defined and I think the manuscript can be accepted for publication as is

Author Response

Dear Sir!

Thank you very much for the analysis of our thesis and the comments made. We have corrected the text in accordance with your comments. Changed parts of the text are highlighted in blue font.

…….I think the manuscript can be accepted for publication as is.

Thank you co very much.